# Microscopic theory, analysis, and interpretation of conductance histograms in molecular junctions

Leopoldo Mejía [1,6,7] ✉, Pilar Cossio[2,3,4] & Ignacio Franco [1,5] ✉

Molecular electronics break-junction experiments are widely used to investigate fundamental physics and chemistry at the nanoscale. Reproducibility in these experiments relies on measuring conductance on thousands of freshly formed molecular junctions, yielding a broad histogram of conductance events. Experiments typically focus on the most probable conductance, while the information content of the conductance histogram has remained unclear. Here we develop a microscopic theory for the conductance histogram by merging the theory of force-spectroscopy with molecular conductance. The procedure yields analytical equations that accurately fit the conductance histogram of a wide range of molecular junctions and augments the information content that can be extracted from them. Our formulation captures contributions to the conductance dispersion due to conductance changes during the mechanical elongation inherent to the experiments. In turn, the histogram shape is determined by the non-equilibrium stochastic features of junction rupture and formation. The microscopic parameters in the theory capture the junction's electromechanical properties and can be isolated from separate conductance and rupture force (or junction-lifetime) measurements. The predicted behavior can be used to test the range of validity of the theory, understand the conductance histograms, design molecular junction experiments with enhanced resolution and molecular devices with more reproducible conductance properties.

The study of charge transport across single-molecules is a powerful tool to investigate fundamental physics and chemistry at the nanoscale[1–7]. In particular, single-molecule conductance measurements have been used to investigate conformational dynamics[8–11], chemical reactions[12–17], quantum interference[18–20], charge transport coherence[21] and to develop single-molecule spectroscopies[22,23]. Further, they are routinely used to establish structure–transport relations that can guide the design and our ability to understand

photovoltaics[24–26], redox catalysis[27], energy transport and storage[28], photosynthesis[29], and biological signaling[30]. In addition, the platform has been used to construct molecular-based devices such as switches[11,31], transistors[32–35], and diodes[36,37].

A common scheme to measure single-molecule conductance is the so-called break-junction setup[38–43]. In these experiments (see Fig. 1) two metallic electrodes are brought into mechanical contact and then pulled apart until a nanoscale gap forms between them due to the

[1]Department of Chemistry, University of Rochester, Rochester, NY 14627, USA. [2]Center for Computational Mathematics, Flatiron Institute, New York City, NY 10010, USA. [3]Center for Computational Biology, Flatiron Institute, New York City, NY 10010, USA. [4]Biophysics of Tropical Diseases Max Planck Tandem Group, University of Antioquia, 050010 Medellín, Colombia. [5]Department of Physics, University of Rochester, Rochester, NY 14627, USA. [6]Present address: Department of Chemistry, University of California, Berkeley, CA 94720, USA. [7]Present address: Materials Sciences Division, Lawrence Berkeley National Laboratory, Berkeley, CA 94720, USA. ✉e-mail: leopoldo.mejia@berkeley.edu; ignacio.franco@rochester.edu

**Fig. 1 | Break-junction experiments.** In these experiments a metallic tip is brought into contact with a metallic surface and the overall length of the junction $L$ is increased by applying an external force $F$ until rupture at an elongation $L_f$. **a** The rupture of the metal-metal junction leads to an initial electrode–electrode gap $\xi_0$ in which the molecule is anchored forming a molecular junction. **b** The pulling of the molecular junction results in its rupture at electrode gap $\xi_r$. Both $\xi_0$ and $\xi_r$ are stochastic variables determined by rupture statistics. The circled numbers in **a**, **b** signal the steps into which the process has been divided, as described in Section "Results". **c** The metal–metal rupture can be seen as the rupture of two brittle springs connected in series. The blue and red springs represent the electrodes (surface and cantilever in a scanning tunneling microscope break-junction experiment). Increasing the overall junction length $L$ from contact to rupture by $\Delta L_f$ leaves an identical junction gap $\xi_0$. **d** The molecular junction rupture can be represented by the rupture of three brittle springs connected in series, where the purple spring represents the molecule.

rupture of the metal–metal junction. Molecules in the surrounding medium bridge the gap between the two electrodes by attaching their ends to the metallic contacts, which results in the formation of a molecular junction. As the formed molecular junction is elongated by mechanical pulling, a voltage is applied and the resulting current is recorded. The pulling is continued until the molecular junction ruptures. This process is repeated thousands of times on freshly formed junctions and the distribution of conductance events, the conductance histogram, is reported.

While the conductance of individual molecular junctions is challenging to experimentally reproduce, the conductance histogram is highly reproducible. Nevertheless, these histograms typically exhibit a broad conductance dispersion of ~0.5–2 orders of magnitude with respect to the most probable conductance value[7,43–47]. This limits the utility of break-junction techniques as a platform to investigate single molecules and construct molecular-based devices, as the broad conductance features impose fundamental limits on the resolution of individual molecular events and the design of devices with reproducible conductance properties.

To extract physical information from the conductance histograms, and design useful strategies to narrow their width, it is desirable to develop a microscopic theory of the conductance distribution in break-junction experiments. Such a theory could be used to interpret and predict the role that external factors—such as the pulling speed, cantilever stiffness, and temperature–, and internal molecular features —such as the molecular structure and chemical anchor groups–, play in determining the width and shape of the conductance histograms. Further, the theory could help bridge the gap between atomistic simulations of molecular conductance that typically focus on few representative junction conformations and measurements that record all statistically possible experimentally accessible events, and thus require a statistical approach[46,48].

However, the complexity of the microscopic origin of the conductance dispersion in molecular junctions has prevented the development of such a theory. Specifically, it has been shown that multiple factors, including changes in the molecular conformation[8,49], various electrode-molecule binding configurations[50,51], variations in the electrode geometry[52,53], and the systematic mechanical manipulation of the junction[44] broaden the conductance histogram, making it challenging to formulate a theory in a unified framework.

To make systematic progress, it is necessary to focus on the contributions of a few microscopic factors to the conductance dispersion. In this regard, a recent theory-experiment analysis[44] revealed that a broad conductance histogram will still emerge even in ideal experiments where the electrode geometry and molecular binding configuration can be perfectly controlled. Specifically, it was observed that the changes in conductance due to the mechanical manipulation of the junction alone already account for the observed conductance dispersion in break-junction experiments. In addition, the study in ref. 44 showed that one of the reasons why these experiments require collecting statistics is because forming and breaking the junction is an inherently stochastic process that needs to be sampled to generate reproducible histograms.

Here, we propose a microscopic theory of conductance histograms by considering the mechanical manipulation of the junction and the statistics of junction formation and rupture as the only sources of conductance dispersion. The theory describes the conductance histograms in terms of physically meaningful parameters that characterize the free-energy profile (FEP) of the junction, its mechanical manipulation, and the ability of the molecule to transport charge. While this view of the origin of the conductance histogram is only capturing one of the possible contributing factors to the conductance dispersion, it allows us to obtain analytic expressions that are useful for fitting, modeling, and interpreting experimental conductance histograms.

This study complements and advances previous efforts to model conductance histograms through phenomenological broadening of junction parameters[54–56] and molecular dynamics simulations of junction formation and evolution[44,46,57], and efforts to classify molecular conductance events through machine learning[7,58,59]. The main advance is that it provides a useful microscopic picture of junction formation and evolution that recovers the shape of experimental conductance histogram, enables extracting information about the microscopic parameters, and establishes a foundation for generalizations.

Specifically, inspired by force-spectroscopy experiments for protein unfolding and unbinding processes[60–62], we develop a theory of break-junction experiments in which the metal-metal and metal-molecule rupture events, that lead to the formation and breaking of the molecular junction, are modeled as stochastic escapes from one-dimensional wells modulated by mechanical forces. This leads to a range of initial and final gaps between electrodes (electrode gap) that determine the molecular ensemble that is sampled in the experiment. By relating the electrode gap to junction conductance, we isolate an expression for the conductance histogram. The main results of this work are analytical equations (Eqs. (6) and (8)) that can properly capture the shape and peak position of experimental conductance histograms—including aliphatic, aromatic, and radical-containing molecules with varying anchor groups, and supramolecular complexes—and that are defined by microscopic parameters that capture the electro-mechanical properties of the junction. The theory can be used to understand how the histograms change with molecular design and experimental conditions such as pulling speed, junction stiffness, and temperature. Importantly, all microscopic features that define the conductance histogram can be extracted, for example, by supplementing the conductance break-junction experiments with force spectroscopy of junction rupture, thus providing a general platform to augment the information content that can be extracted from this class of experiments.

## Results

To develop a theory for the conductance histograms, we partition the break-junction experiment (Fig. 1) into six main events: (1) the formation of a contact between the two metallic electrodes (Fig. 1a, left); (2) the mechanical elongation of the metal-metal contact and (3) its rupture to create a nanoscopic gap (Fig. 1a, right); (4) the attachment of a molecule bridging this gap between the two electrodes that is (5) subsequently mechanically elongated (Fig. 1b, left) until (6) junction rupture (Fig. 1b, right). Processes (3) and (6) are stochastic, thermally activated, and nonequilibrium in nature. Our view is that, because of this, to recover reproducible conductance features it is necessary to statistically sample all possible rupture events by repeating the experiment (steps (1)–(6)) thousands of times.

In the theory, the conductances that enter into the histograms are those encountered by the junction during (5). The distribution of junction elongations that determine such conductances are given by the distribution of initial nanoscopic gaps in (3) and at rupture in (6). The probability of a given conductance value, thus, depends on the probability that a given electrode gap is visited during (5) and the relation between molecular junction conductance and such electrode gaps.

The theory supposes that there is an effective one-to-one relation between measured conductance $\langle G(\xi) \rangle$ and junction gap $\xi$. For a given gap, there is a whole thermal ensemble of possible molecular and junction configurations that are accessible and contribute to the conductance. However, since experiments measure a current that is time-averaged over microseconds, these individual contributions are averaged out and cannot be experimentally resolved leading to simpler conductance traces. One coordinate that systematically changes during pulling in timescales slower than the integration time of the current is the electrode gap, $\xi$. In experiments, the junction is pulled

with speeds of nm/s, and thus sub-Ångstrom variations of electrode gap $\xi$ can be experimentally resolved in the conductance measurements[44]. This effectively leads to a conductance that parametrically depends on the electrode gap, $\langle G(\xi) \rangle$.

We note that, even in the presence of time-averaging, the experiments can discriminate molecular conformations that are mechanically stabilized, or that survive for times longer than the current integration time, such as changes in the binding configuration or transitions between stable molecular conformations (e.g., gauche vs. trans isomers in alkanes or mechanically activated reactions). The one-to-one assumption for $\langle G(\xi) \rangle$ accounts for many of these changes, but it cannot capture physical situations in which multimodal conductance features are accessible at a given electrode gap. When there is no interconversion between these modes, multimodality can be simply accounted for through independent one-to-one $\langle G(\xi) \rangle$ relations. The proposed theory can also be generalized to the more complex case where different long-lived conformers are accessible at a given $\xi$ and interconvert during the pulling process. For definiteness, we focus on a one-to-one relation for $\langle G(\xi) \rangle$ and discuss its generalization in the outlook.

The experimentally controllable variable is the overall length of the junction $L$ and not $\xi$ (Fig. 1). These quantities do not coincide as during pulling there can be mechanical deformations of the electrodes that do not lead to changes in the electrode gap. In experiments, it is customary to align different conductance traces $\langle G(L) \rangle$ at their rupture point. In ref. 44 we showed computationally that even in ideal experiments (with reproducible electrode shape and binding configuration) this will lead to a dispersion of $\langle G(L) \rangle$ curves as the elongation in which the junction ruptures varies between experiments. However, as shown in ref. 44, if the conductance trajectories are plotted with respect to the electrode gap they collapse into a single curve, justifying the one-to-one relation $\langle G(\xi) \rangle$ in the theory.

To describe the nonequilibrium stochastic features inherent to the junction formation and rupture, we take advantage of the theory of rupture-force statistics developed in biophysics to describe mechanically modulated transitions between two states[60–65] as needed, for instance, in the description of the force-spectroscopy of protein unfolding. To map this into conductance, which is the main observable, we first develop a model for the mechanical driving of the molecular junction that maps the distribution of rupture forces into a distribution of electrode gaps at junction formation and rupture. Then, based on the parametric relation between the average conductance and electrode gap $\langle G(\xi) \rangle$, we connect the distribution of initial and final junction elongations to a distribution of conductances. This results in a general equation for the conductance histogram. We further specialize the model to the case in which the conductance depends exponentially on $\xi$ which is then used to fit representative experiments and analyze the theory. For clarity in presentation, Table 1 summarizes the symbols and physical meaning of the parameters of the theory.

### The probability density function of rupture-forces

The rupture of the metal–metal contact and of the molecular junction (processes (3) and (6)) can be understood as a free energy barrier-crossing event in the presence of an external force from an unruptured state to a ruptured state. Specifically, the thermodynamic state of the (metal-metal or molecular) junction is represented by a one-dimensional FEP along the pulling coordinate (see Fig. 2). The unruptured state corresponds to the bottom-well in the FEP with equilibrium elongation $\xi_{eq}$ at zero force ($F = 0$). The transition state between the ruptured and unruptured state, represented by the barrier top, is located a distance $\chi^{\ddagger}$ away from the well bottom, and has a $\Delta A_0^{\ddagger}$ activation free energy at $F = 0$. The unruptured state corresponds to the unbounded portion of the FEP ($\xi > \xi_{eq} + \chi^{\ddagger}$).

The application of an external force modifies the FEP, modulating the junction rupture rates. For example, positive forces ($F > 0$) reduce

**Table 1 | Physical parameters defining the theory of conductance histograms**

| Symbol | Meaning | Example | |
|---|---|---|---|
| $\beta$ | Inverse temperature | 38.68 eV$^{-1}$ (300 K) | |
| $\dot{F}$ | Loading rate | 400 nN s$^{-1}$ (f) | |
| | | 400 nN s$^{-1}$ (r) | |
| $\kappa$ | Junction spring constant | 8 N m$^{-1}$ (f) | |
| | | 8 N m$^{-1}$ (r) | |
| $\kappa_\xi$ | Molecular spring constant | 8 N m$^{-1}$ | |
| | | | Fitted (SD) |
| $\xi_{eq}$ | Molecular junction electrode gap at mechanical equilibrium | 1.50 Å | 1.35 (0.05) Å |
| $\chi^\ddagger$ | Distance between $\xi_{eq}$ and transition state in the FEP | 0.200 Å (f) | 0.198 (0.002) Å (f) |
| | | 0.200 Å (r) | 0.199 (0.002) Å (r) |
| $k_0$ | Spontaneous rupture rate ($F = 0$) | 1.00 s$^{-1}$ (f) | 1.30 (0.08) s$^{-1}$ (f) |
| | | 20.00 s$^{-1}$ (r) | 24.81 (1.12) s$^{-1}$ (r) |
| $\gamma$ | Transmission decay coefficient | −1.15 Å$^{-1}$ | −1.14 (0.01) Å$^{-1}$ |
| $\log T_0$ | Base transmission ($\xi \to 0$) | −3.00 | −2.97 (0.02) |

Example values (column 3) were used to construct Figs. 3 and 4. Parameters in column 4 were recovered from fitting synthetic rupture force and conductance histograms, generated with the example parameters in column 3, to Eqs. (2) and (8) (standard deviation in brackets). The (f) and (r) symbols in brackets refer to the rupture of the metal-metal and molecular junction, respectively. The quantities $\beta$, $\dot{F}$, $\kappa$ and $\kappa_\xi$ are experimentally known. All other parameters can be extracted by fitting experimental data to the theory.

the activation barrier making it more probable for the junction to rupture. We assume a brittle system where $\xi_{eq}$ and the distance to the transition state ($\chi^\ddagger$) do not change with force as shown in Fig. 2. In this mechanically brittle limit[60], the force-dependent rupture rate follows Bell's formula[65]:

$$k(t) = k_0 e^{\beta\chi^\ddagger F(t)}, \qquad (1)$$

where $k_0$ is the spontaneous rupture rate at $F = 0$, $\beta$ is the inverse temperature and where the force $F(t) = \dot{F}t$ is assumed to increase linearly in time with a constant loading rate $\dot{F}$. Equation (1) implies that the activation energy $\Delta A^\ddagger = \Delta A_0^\ddagger - F(t)\chi^\ddagger$ varies linearly with $F(t)$, as represented in Fig. 2 for positive (pulling) and negative (pushing) forces. This was shown by Bell[65] for a soft spring constant $\kappa$ by considering that the FEP is distorted in the presence of forces as $A(\xi, t) = A_0(\xi) + \kappa(\xi - (\dot{F}/\kappa)t)^2/2$ and using Arrhenius formula. Under the assumption that the survival probability of the junction $S(t)$ follows a first-order rate equation of the form $\dot{S}(t) = -k(t)S(t)$ and using Bell's formula (Eq. (1)) for the rate coefficient, the probability density function of rupture forces $p_F(F)$−i.e., the force required to mechanically break the junction−is (see Ref. 61 for details):

$$p_F(F) = \frac{k_0}{\dot{F}} \exp\left[\beta\chi^\ddagger F - \frac{k_0}{\dot{F}\beta\chi^\ddagger} e^{\beta\chi^\ddagger F}\right]. \qquad (2)$$

Equation (2) was first obtained in Ref. 66 to describe the dynamics of the unbinding of the Avidin-Biotin complex, and has been widely used to investigate the kinetics of single-molecule pulling experiments in the context of biophysics[61,65,67–71]. Its accuracy depends on the range of validity of Bell's approximation. It has been found that Bell's approach is accurate in the low-force regime[72] where the applied force does not completely deplete the activation free-energy barrier.

In the context of molecular electronics scanning tunneling microscopy break-junction (STM-BJ) experiments, the rupture force of

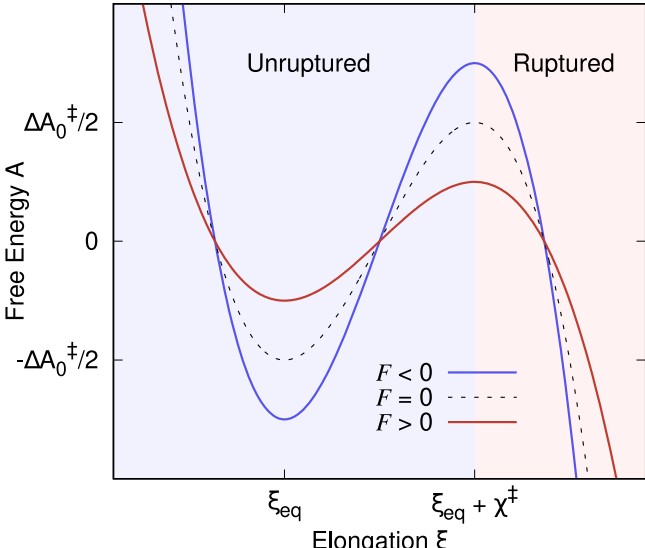

**Fig. 2 | Schematic representation of the free-energy profile (FEP) of a metal-metal or molecular junction along the pulling coordinate.** Here $\xi$ is the electrode gap and $\xi_{eq}$ its value at mechanical equilibrium. External mechanical forces ($F$) decrease (red line, $F > 0$) or increase (blue line, $F < 0$) the free-energy barrier $\Delta A^\ddagger$ (which is located a distance $\chi^\ddagger$ away from $\xi_{eq}$) between the unruptured and ruptured states with respect to that of the pristine junction ($\Delta A_0^\ddagger$).

a metal-metal[73] and molecular[74] junction has well-defined experimentally-accessible regimes where it satisfies Bell's formula. Both junctions also show a regime where the rupture force becomes independent of pulling rate that is beyond the regime of applicability of the theory. Our analysis pertains to experiments performed under conditions in which Bell's theory is applicable.

### Molecular junction gaps along pulling

Break-junction experiments involve two junction rupture events during the mechanical pulling: the rupture of the metal-metal contact to create the gap in which the molecule is initially placed (Fig. 1a) and the rupture of the formed molecular junction (Fig. 1b). Each of these rupture events has an associated probability density function of rupture forces determined by Eq. (2).

As represented in Fig. 1c, the pulling of the metal−metal junction is analogous to the pulling of two brittle springs connected in series. The bottom spring (red in Fig. 1c) represents the deformation of the bottom electrode surface with elasticity constant $\kappa_{surf}$. The top spring (blue in Fig. 1c) represents the deformation of a cantilever or top electrode (whichever is softer) with spring constant $\kappa_{cant}$. The effective spring constant of the composite system ($\kappa_f$) is then given by $1/\kappa_f = 1/\kappa_{surf} + 1/\kappa_{cant}$.

The junction is elongated $\Delta L > 0$ until its rupture at $\Delta L = \Delta L_f$. At this point the electrodes return to their mechanical equilibrium leaving a gap $\Delta L_f = \xi_0$, where we have assumed that there is no plastic deformation of the gold electrodes, i.e., changes in the electrode geometry due to the mechanical manipulation[75–78]. Such events change the equilibrium length of the electrodes but leave the analysis intact. Further extensions of the model that allow directly estimating the free energy barrier[79] and transitions from brittle to ductile regimes can be captured by adding an additional parameter to the free energy profile, as proposed in ref. 60.

The distribution of metal-metal rupture forces determines the distribution of initial $\xi_0$ gaps. Specifically, a rupture force $F_f = \kappa_f\Delta L_f = \kappa_f\xi_0$ leads to a $\xi_0 = \frac{F_f}{\kappa_f}$ gap. Therefore, the probability density function of initial electrode-electrode gaps, $p_0(\xi_0)$, can be calculated from the probability density function of rupture forces $p_F(F)$, as

$p_0(\xi_0) = \kappa_f p_F(F = \kappa_f \xi_0)$. Then, from Eq. (2) it follows that

$$p_0(\xi_0) = \frac{k_{0f}\kappa_f}{\dot{F}_f} \exp\left[\beta\chi_f^\ddagger \kappa_f \xi_0 - \frac{k_{0f}}{\dot{F}_f \beta\chi_f^\ddagger} e^{\beta\chi_f^\ddagger \kappa_f \xi_0}\right]. \quad (3)$$

All quantities in Eq. (3) refer to the metal–metal rupture and its FEP, as signaled by the "f" (junction formation) subscript throughout.

Similarly, the probability density function of molecular junction gaps at rupture ($\xi_r$) is determined by the probability density function of rupture forces of the molecular junction. As proposed in Fig. 1d, the mechanical response of the molecular junction can be viewed as three brittle springs connected in series. When a force is applied to the combined spring, the same force is applied to each individual spring. Thus, the exerted force at rupture $F_r = \kappa_r \Delta L_r$, where $\kappa_r$ is the overall spring constant and $\Delta L$ the overall elongation, can be written in terms of the gap between electrodes $\Delta\xi$ as $F_r = \kappa_\xi \Delta\xi = \kappa_\xi(\xi_r - \xi_{eq})$, where $\kappa_\xi$ is the molecular effective spring constant and $\xi_{eq}$ is the electrode gap at which the molecular junction is in mechanical equilibrium ($F = 0$). This yields an expression for the probability density function of electrode gaps at rupture given by

$$p_r(\xi_r) = \frac{k_{0r}\kappa_\xi}{\dot{F}_r} \exp\left[\beta\chi_r^\ddagger \kappa_\xi(\xi_r - \xi_{eq}) - \frac{k_{0r}}{\dot{F}_r \beta\chi_r^\ddagger} e^{\beta\chi_r^\ddagger \kappa_\xi(\xi_r - \xi_{eq})}\right]. \quad (4)$$

Here, all quantities refer to molecular junction rupture and its associated FEP as signaled by the "r" (junction rupture) subscript throughout. Figure 3a shows the $p_0(\xi_0)$ and $p_r(\xi_r)$ probability density functions for a model system with the set of example parameters shown in Table 1. These parameters were chosen to have values that are representative of break-junction experiments of gold–gold contacts and gold–alkane–gold molecular junctions[73,74]. As discussed below, the resulting rupture force and conductance histograms obtained from these parameters are within the range of values that are typically measured in experiments.

The probability density function of initial ($p_0(\xi_0)$) and final ($p_r(\xi_r)$) molecular elongations during pulling determines the probability of visiting a particular molecular junction gap, $\xi$, during the break-junction experiment. To extract this quantity, we assume that for a given trajectory all $\xi$ are equally probable between a given initial and rupture points, in agreement with the constant pulling speed in these

experiments and the harmonic picture for the mechanical deformation. The probability of visiting a particular electrode gap during pulling corresponds to the product of the probabilities that the junction has been formed already ($P_0(\xi)$) and has not been ruptured ($1 - P_r(\xi)$) at $\xi$. Therefore, the probability of sampling an electrode gap $\xi$ is

$$P(\xi) = \overbrace{\left(\int_{-\infty}^{\xi} p_0(\xi_0)d\xi_0\right)}^{P_0(\xi)} \overbrace{\left(1 - \int_{-\infty}^{\xi} p_r(\xi_r)d\xi_r\right)}^{1-P_r(\xi)}$$
$$= \left(1 - \exp\left[-\frac{k_{0f}}{\dot{F}_f \beta\chi_f^\ddagger} e^{\beta\chi_f^\ddagger \kappa_f \xi}\right]\right) \quad (5)$$
$$\times \exp\left[-\frac{k_{0r}}{\dot{F}_r \beta\chi_r^\ddagger} e^{\beta\chi_r^\ddagger \kappa_\xi(\xi - \xi_{eq})}\right].$$

We considered the application of forces in the $(-\infty, \infty)$ range to account for the pushing and pulling of the junction. Figure 3b shows the resulting $P(\xi)$ and its contributions from the probabilities of junction formation and rupture (dashed lines) calculated with Eq. (5) for the parameters in Table 1.

## Emergence of conductance histograms

Equation (5) gives the probability that a given electrode gap is sampled in a break-junction experiment. To recover the conductance histograms, we connect $P(\xi)$ to the probability density of measuring a time-averaged log-transmission $\log T = \log(\langle G(\xi)\rangle/G_0) \equiv g(\xi)$ (where $G_0 = \frac{2e^2}{h}$ is the quantum of conductance) by taking advantage of the (one-to-one) relation between electrode gap and junction conductance. Therefore, Eq. (5) can be rewritten as a probability density for $\log T$ using $\xi = g^{-1}(\log T)$

$$p(\log T) = N\left(1 - \exp\left[-\frac{k_{0f}}{\dot{F}_f \beta\chi_f^\ddagger} e^{\beta\chi_f^\ddagger \kappa_f g^{-1}(\log T)}\right]\right) \times$$
$$\exp\left[-\frac{k_{0r}}{\dot{F}_r \beta\chi_r^\ddagger} e^{\beta\chi_r^\ddagger \kappa_\xi(g^{-1}(\log T) - g^{-1}(\log T_{eq}))}\right], \quad (6)$$

where $\log T_{eq} = g(\xi_{eq})$ and $N$ is a normalization constant. Equation (6) is an analytical expression for the conductance lineshape in break-junction experiments with a generic dependence between the time-

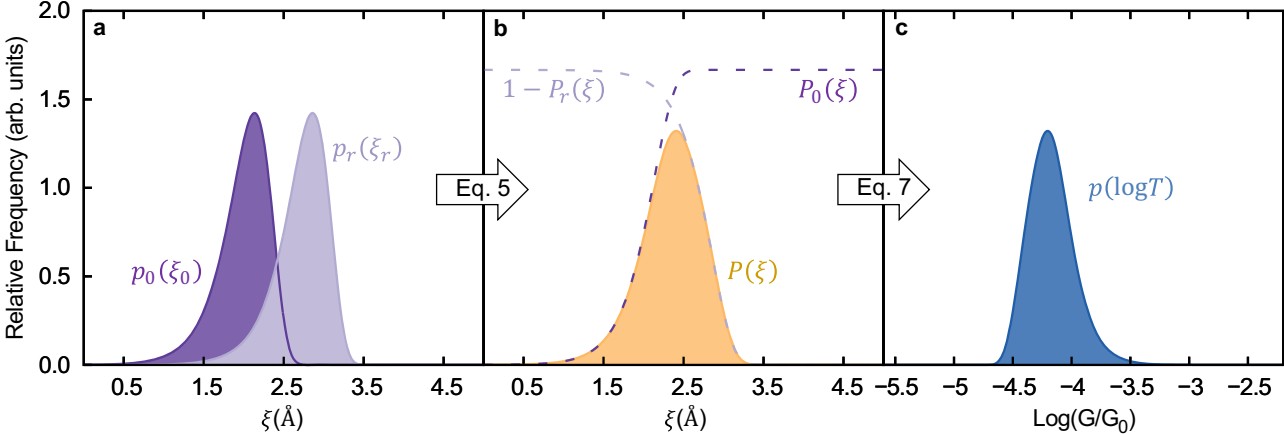

**Fig. 3 | Modeling of conductance histogram in a break-junction experiment.** **a** Probability density function of the initial ($p_0(\xi_0)$, Eq. (3)) and rupture ($p_r(\xi_r)$, Eq. (4)) electrode gaps $\xi$ in a molecular junction. **b** Probability of visiting the electrode gap $\xi$ ($P(\xi)$) during a break-junction experiment. The dotted lines represent the probability that the junction has been formed ($P_0(\xi)$)/has not been ruptured ($1 - P_r(\xi)$) at a given $\xi$. **c** Conductance $G$ histogram calculated with Eq. (8). The histogram is reported on a log scale and in units of the quantum of conductance $G_0 = 2e^2/h$. In all cases, the parameters in Table 1 were used.

averaged conductance and electrode gap. If the experiments have featureless additive background noise, such as conductance contributions coming from direct electrode–electrode tunneling[45,54,80], then $p(\log T)_{EXP} = N_1 p(\log T) + N_0$ where $N_1/N_0$ can be understood as the signal to noise ratio.

Computing a conductance histogram using Eq. (6) requires a specific form for $g(\xi)$. We now specialize our considerations to the case in which the average transmission $T = T_0 e^{\gamma \xi}$ exponentially increases ($\gamma > 0$) or decreases ($\gamma < 0$) with the electrode gap. In this case,

$$g(\xi) = \log T = \frac{\gamma}{\ln 10} \xi + \log T_0 \tag{7}$$

is a linear function of $\xi$ with slope determined by $\gamma$. Here, the base transmission $T_0$ is defined as the extrapolated transmission at $\xi \to 0$. This functional dependence was observed in detailed atomistic simulations of alkane-based junctions[44]. As discussed below, this specific $g(\xi)$ enables the development of a tractable theory and captures the conductance histograms of a wide class of molecules. Other possible forms can be developed to capture additional features of the conductance histograms.

Substituting Eq. (7) in Eq. (6) results in an expression for the probability density function of $\log T$

$$p(\log T) = N \left(1 - \exp\left[-c_2 e^{c_1 \log T}\right]\right) \exp\left[-c_4 e^{c_3 \log T}\right]. \tag{8}$$

where $c_{1,2}$ and $c_{3,4}$ are characteristic coefficients due to the molecular-junction formation and rupture, respectively, given by

$$c_1 = \frac{\beta \chi_f^{\ddagger} \kappa_f}{\gamma} \ln 10, \tag{9}$$

$$c_2 = \frac{k_{0f}}{\dot{F}_f \beta \chi_f^{\ddagger}} e^{-c_1 \log T_0}, \tag{10}$$

$$c_3 = \frac{\beta \chi_r^{\ddagger} \kappa_\xi}{\gamma} \ln 10, \tag{11}$$

and

$$c_4 = \frac{k_{0r}}{\dot{F}_r \beta \chi_r^{\ddagger}} e^{-c_3 \log T_{eq}}. \tag{12}$$

Here, $\log T_{eq} = \frac{\gamma}{\ln 10} \xi_{eq} + \log T_0$ is the log-transmission at the equilibrium electrode gap. Figure 3c shows the resulting conductance histogram calculated with Eq. (8) for the parameters in Table 1. Equations (6) and (8) are the main results of this section.

## Discussion

Equation (8) provides an analytical expression for the break-junction conductance histograms in terms of physically meaningful parameters. In particular, the coefficients $c_1$ and $c_2$ (Eqs. (9) and (10), respectively) capture the contributions from the metal-metal rupture kinetics that precede the formation of the molecular junction to the conductance histogram. In turn, parameters $c_3$ and $c_4$ (Eqs. (11) and (12), respectively) capture the contributions from the molecular-junction rupture process. These four coefficients are defined by microscopic parameters describing the free-energy profile of the junction, its mechanical manipulation, and the ability of the molecule to transport charge (see Table 1 for definitions). For completeness, Fig. S1 in the Supporting Information illustrates the effect of independently varying each coefficient $c_1$–$c_4$ on the conductance histogram.

### Effect of the microscopic parameters on the conductance histogram

Equation (8) enables to elucidate the effects of the microscopic parameters on the conductance histogram. For example, Fig. 4 shows the effect of varying $T_0$, $\gamma$, and $\dot{F}$ (the influence of the remaining parameters is included in Fig. S2). The transmission decay coefficient $\gamma$ and the base transmission $T_0$ are the quantities that define the intrinsic transport properties of the molecule. Figure 4a shows that $\log T_0$ displaces the distribution of conductances without changing its shape in the logarithmic scale. By contrast, Fig. 4b shows that the width of the histogram is determined by $\gamma$. Specifically, small values of $|\gamma|$ correspond to molecular junctions whose conductance is not very sensitive to changes in elongation, resulting in narrow histograms. In the figure, we have covered a wide range of values of $\gamma$ to exemplify this effect; actual experiments might show less dramatic changes. Nevertheless, the chemical design of molecules with small $|\gamma|$ is the key to creating molecular junctions with reproducible conductance features.

Note that the probability distribution of $\xi$ visited once the junction is formed, $P(\xi)$, is asymmetric with a tail toward smaller $\xi$ (see Eq. (5) and Fig. 3). This asymmetry leads to a tail in the conductance histogram. In our model, negative/positive values of $\gamma$ lead to conductance tails towards the higher/lower conductance values, respectively. Possible additional asymmetries in the conductance

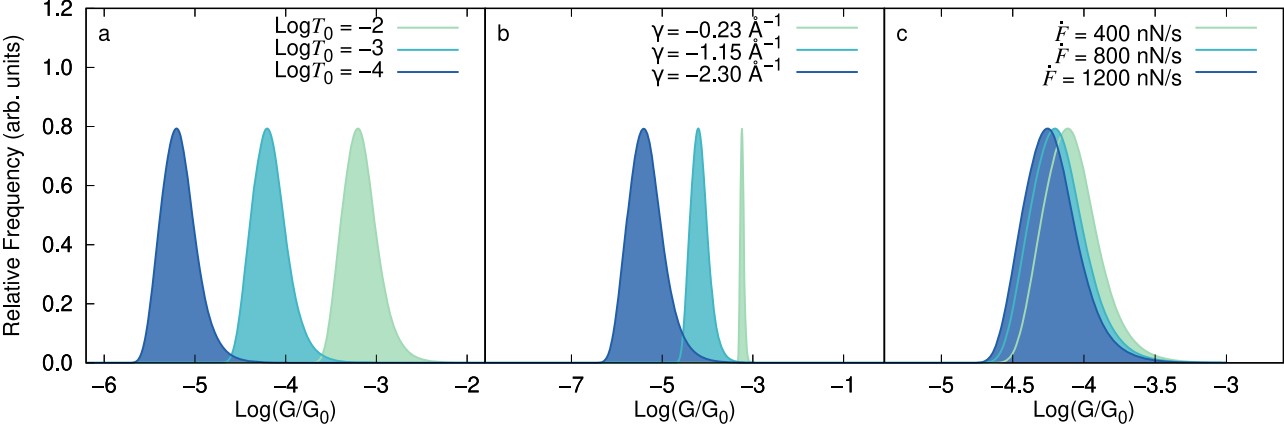

**Fig. 4 | Influence of the microscopic parameters and loading rate on the conductance $G$ histogram. a** Effect of the base transmission $T_0$, **b** transmission decay coefficient $\gamma$, and **c** loading rate $\dot{F}$ on the break-junction conductance histograms, calculated with Eq. (8). For these cases, all parameters except the one being varied are those in Table 1. The histograms are reported on a log scale and in units of the quantum of conductance $G_0$. The influence of the remaining microscopic parameters is included in Fig. S2.

histogram that are introduced by background noise in the experiments[45,80,81] need to be removed before inferring the sign of $\gamma$ from experimental data.

The loading rate, $\dot{F} = \kappa\nu$, is proportional to the pulling speed $\nu$ and effective spring constant of the junction $\kappa$ ($\kappa = \kappa_{f/r}$). Figure 4c shows that for a molecule with $\gamma < 0$, decreasing the loading rate results in the conductance histogram shifting toward higher conductance values. This is because when the junction is elongated slowly, statistically, it breaks at shorter elongations. For $\gamma > 0$, decreasing $\dot{F}$ shifts the histograms toward lower conductance values.

The exponential (or any other monotonic) relation between the electrode gap and conductance will result in histograms that depend on the loading rate. This dependency has not been experimentally observed yet in the few characterizations that have been conducted[74,82,83]. Under the experimentally realistic conditions of Table 1, the conductance histogram changes only slightly with loading rate. In fact, varying the loading rate from 400 to 1200 nN/s only shifts the conductance peak from $\log(\langle G \rangle/G_0) = -4.11$ to $-4.25$ (about 0.8 standard deviations of the (log) conductance histogram) which may be challenging to resolve. Therefore, an important challenge for

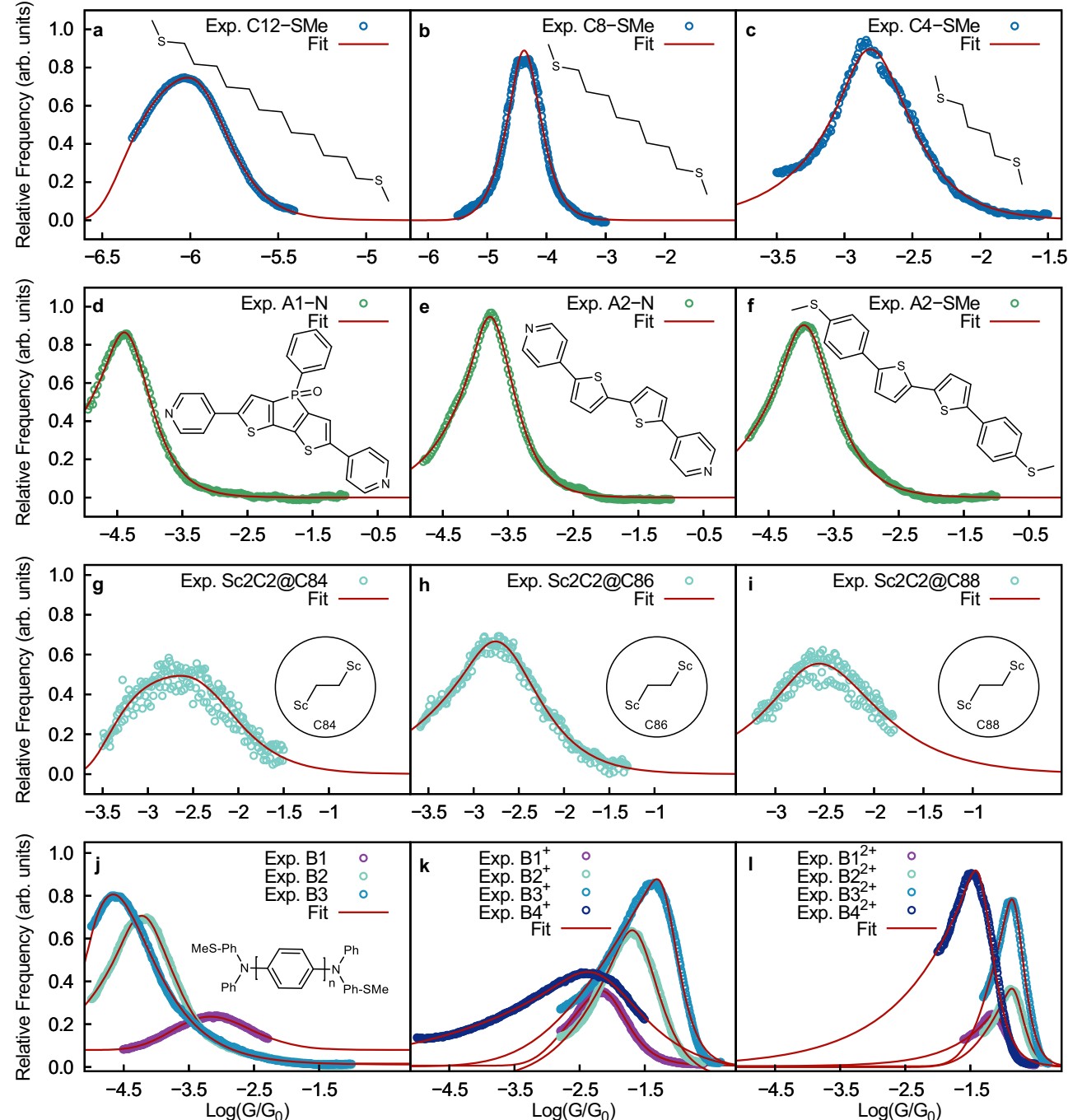

**Fig. 5 | Utility of Eq. (8) in fitting experimental conductance $G$ histograms.** The plot shows experimental conductance histograms of junctions formed with **a**–**c** aliphatic molecules (C$n$-SMe), **d**–**f** aromatic molecules (A$n$-(N or SMe)), **g**–**i** metallofullerene complexes (Sc2C2@C$n$), **j**–**l** radical containing molecules of varying length and charge (B$n^{m+}$) and their accurate fit to Eq. (8). The values of the fitting parameters are shown in Table 2. The histograms are reported on a log scale and in units of the quantum of conductance $G_0$. Experimental data was provided by Prof. Venkataraman for C$n$-SMe and obtained from refs. 86–88 for the other cases.

future experiments is to better characterize the dependence of the histograms on loading rate to determine if non-monotonic conductance-electrode gap relations are required to better understand the conductance histograms.

**Fit to experimental conductance histograms**

To demonstrate that Eq. (8) is useful in analyzing experimental data, we tested its ability to fit STM conductance histograms for a wide variety of molecular junctions. Figure 5 shows the experimental conductance histograms of twenty representative molecular junctions and their excellent fit to Eq. (8). The extracted parameters are included in Table 2. Even when this is a highly nonlinear fit, we observe that the parameters extracted are robust (see Fig. S3 and Table S1 in the supplementary information). The set includes junctions formed with (a)-(c) aliphatic (SMe)-$(CH)_n$-(SMe)-Au molecules ($Cn$SMe), (d)−(f) complex aromatic systems (A1 and A2), (g)−(i) supramolecular complexes composed of metallofullerenes and (j)−(l) radical containing molecules of varying length and charge. Overall, Fig. 5 demonstrates the general utility of Eq. (8) to fit experimental histograms.

For completeness, in the supplementary information, we compare the fits using our microscopic theory with the phenomenological approach by Reuter and Ratner in refs. 54,56 based on Gaussian broadening of energy levels and molecule-metal couplings. While both show good agreement (Fig. S4 and Table S2 in the Supporting Information), the fits to Eq. (8) are statistically better. Furthermore, our theory has the advantage of being based on a microscopic model that can be used to make physical predictions and advance molecular design.

Equation (8) can also be used to identify individual contributions to multimodal conductance histograms. Figure 6 shows the experimental conductance histogram of two representative Au-S-$(CH)_n$-S-Au

junctions ($Cn$-DT). This class of molecular junctions leads to bimodal conductance distributions, corresponding to two stable binding configurations of the thiol anchor group on the Au electrode surface (top and bridge configurations)[44]. We show that the experimental histogram can be fitted to $p_1(\log T) + p_2(\log T)$, where each of these distributions in described by Eq. (8). From this fit, individual low and high conductance peaks are then easily identified, as shown in Fig. 6. The resulting fitting parameters are included in Table 2.

Figures 5 and 6 further demonstrate that Eq. (8) can be used to capture histograms obtained with both donor-acceptor (-SMe-Au and -N-Au) and covalent (-S-Au) anchoring between the molecule and electrodes.

Figures 5 and 6 are based on STM-BJ measurements. Another experimental setup often used to construct conductance histograms is the mechanically-controlled break-junction setup (MCBJ) in a three-point bending configuration[84]. For completeness, we have included fits using Eq. (8) of MCBJ experiments performed on alkanedithiols and alkanediamines[59] in Fig. S5 and Table S2 of the supplementary information. The excellent fits suggest that the functional form is also applicable in this case. However, we do not expect the microscopic parameters needed to define the MCBJ to be identical to those of our model and the STM-BJ, as in the MCBJ the applied force is not parallel to the junction elongation.

**Table 2 | Parameters describing the experimental conductance histograms in Figs. 5 and 6 obtained by fitting Eq. (8) and $R^2$ quality of the fit**

| Molecule | $c_1$ | $c_2$ | $c_3$ | $c_4$ | $R^2$ |
|---|---|---|---|---|---|
| C12-SMe | −7.16 | $7.82 \times 10^{-19}$ | −7.04 | $3.07 \times 10^{-20}$ | 0.998 |
| C8-SMe | −5.72 | $2.18 \times 10^{-11}$ | −1.08 | $1.88 \times 10^{-2}$ | 0.974 |
| C4-SMe | −6.35 | $2.77 \times 10^{-8}$ | −0.01 | $2.52 \times 10^{2}$ | 0.966 |
| A1-N | −4.32 | $1.22 \times 10^{-8}$ | $-2.86 \times 10^{-3}$ | $4.39 \times 10^{2}$ | 0.998 |
| A2-N | −4.84 | $2.15 \times 10^{-8}$ | $-3.72 \times 10^{-3}$ | $4.39 \times 10^{2}$ | 0.996 |
| A2-SMe | −3.95 | $2.82 \times 10^{-8}$ | $-3.26 \times 10^{-3}$ | $4.39 \times 10^{2}$ | 0.992 |
| Sc2C2@C84 | −2.84 | $2.44 \times 10^{-3}$ | −4.22 | $4.93 \times 10^{-7}$ | 0.886 |
| Sc2C2@C86 | −3.60 | $9.41 \times 10^{-5}$ | −0.29 | 1.77 | 0.987 |
| Sc2C2@C88 | −3.22 | $3.63 \times 10^{-4}$ | $-7.34 \times 10^{-2}$ | $1.71 \times 10^{1}$ | 0.812 |
| B1 | −5.60 | $5.06 \times 10^{-12}$ | $-5.02 \times 10^{-1}$ | 2.27 | 0.994 |
| B2 | −3.55 | $5.48 \times 10^{-7}$ | $-9.94 \times 10^{-3}$ | $1.23 \times 10^{2}$ | 0.998 |
| B3 | −1.74 | $6.04 \times 10^{-7}$ | −2.21 | $2.69 \times 10^{-5}$ | 0.999 |
| B1$^+$ | −3.94 | $5.25 \times 10^{-4}$ | −1.81 | $8.51 \times 10^{-3}$ | 0.999 |
| B2$^+$ | −4.10 | $2.21 \times 10^{-3}$ | $-4.948 \times 10^{-1}$ | $9.23 \times 10^{-1}$ | 0.999 |
| B3$^+$ | −5.36 | $3.04 \times 10^{-1}$ | $-5.795 \times 10^{-1}$ | $4.72 \times 10^{-1}$ | 0.992 |
| B4$^+$ | −2.07 | $1.76 \times 10^{-2}$ | $-3.28 \times 10^{-3}$ | $1.49 \times 10^{2}$ | 0.999 |
| B1$^{2+}$ | −8.64 | $1.50 \times 10^{-4}$ | $-5.24 \times 10^{-3}$ | $3.08 \times 10^{2}$ | 0.994 |
| B2$^{2+}$ | −9.01 | $1.16 \times 10^{-3}$ | −1.19 | $5.03 \times 10^{-1}$ | 0.998 |
| B3$^{2+}$ | −8.71 | $1.47 \times 10^{-3}$ | −1.49 | $0.31 \times 10^{1}$ | 0.999 |
| B4$^{2+}$ | −6.38 | $5.04 \times 10^{-4}$ | −2.67 | $3.63 \times 10^{-3}$ | 0.994 |
| C10-DT (low) | −4.53 | $9.58 \times 10^{-12}$ | −1.37 | $4.48 \times 10^{-4}$ | 0.995 |
| C10-DT (high) | −3.08 | $4.31 \times 10^{-10}$ | −3.66 | $6.23 \times 10^{-9}$ | 0.995 |
| C6-DT (low) | −4.53 | $1.87 \times 10^{-9}$ | −0.59 | $3.74 \times 10^{-1}$ | 0.999 |
| C6-DT (high) | −5.05 | $1.58 \times 10^{-7}$ | −3.35 | $3.35 \times 10^{-6}$ | 0.999 |

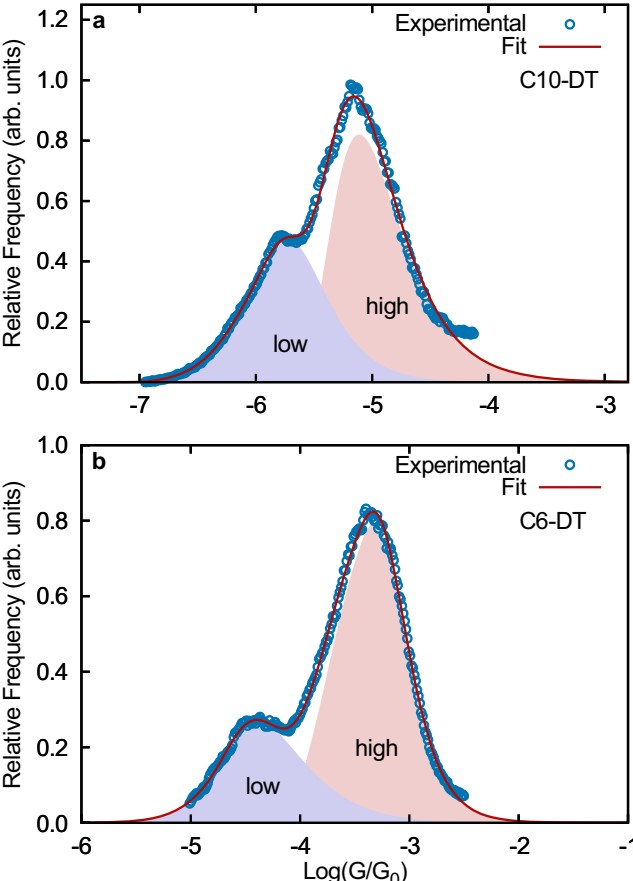

**Fig. 6 | Bimodal experimental conductance $G$ histograms corresponding to the Au-C$n$-DT-Au junctions and their fit to the $p_1(\log T) + p_2(\log T)$ equation.** Here, both $p_1(\log T)$ (in blue) and $p_2(\log T)$ (in red) correspond to Eq. (8) with different fitting parameters shown in Table 2. The fitting of bimodal distributions allows us to identify individual high/low transmission peaks. The histograms are reported on a log scale and in units of the quantum of conductance $G_0$. The experimental data was provided by Professor Latha Venkataraman.

Generally, interpreting $c_1$-$c_4$ directly is challenging since they combine the electrical and mechanical properties of the junction. Below, we describe how to extract the physically interpretable parameters that define the model by supplementing the conductance histograms with force spectroscopy.

### Extracting microscopic parameters

To interpret the conductance histograms it is desirable to extract all microscopic parameters that define the $c_1$-$c_4$ coefficients. One possible strategy to do so is to complement the conductance measurements with force-spectroscopy of both the metal-metal and the molecular junction. In experiments, the inverse temperature $\beta$, loading rates ($\dot{F}_f, \dot{F}_r$) and the elasticity of the junction ($\kappa_f, \kappa_\xi$) are known. Fitting the rupture-force histogram of both the metal-metal and molecular junction to Eq. (2), yields the spontaneous rupture rate ($k_0$) and the distance to the transition state ($\chi^\ddagger$) in each case. Fitting the conductance histogram to Eq. (8) to extract $c_1$-$c_4$, and then using the extracted values for the mechanical parameters in Eqs. (9)–(12) yields the conductance decay coefficient ($\gamma$), the molecular base transmission ($\log T_0$), the transmission at mechanical equilibrium ($\log T_{eq}$) and its corresponding electrode gap ($\xi_{eq}$). This set of parameters completely defines the electro-mechanical model.

To demonstrate this procedure and test its numerical robustness, we generated synthetic rupture force and conductance histograms consistent with the parameters in Table 1 (see SI and Fig. S3 for details). The extracted microscopic parameters from the synthetic data (column 4, Table 1) are in excellent agreement with the original set demonstrating the numerical robustness of the approach.

Another possible strategy is to measure junction lifetimes to extract $k_0$ and $\chi^\ddagger$[85], instead of performing separate force measurements.

In conclusion, we developed a rigorous microscopic theory of conductance histograms in molecular electronics by merging the theory of force-spectroscopy with molecular conductance. As a result, we obtained a general and analytical expression (Eq. (6)) for the break-junction conductance histograms with physically meaningful fitting parameters. Assuming an exponential dependence between the transmission coefficient and electrode gap (Eq. (8)), we obtain practical expressions that provide excellent fits to experimental conductance histograms. The analytical expression has been successfully applied (see Figs. 5 and 6) to a wide variety of molecules including aliphatic, aromatic, supramolecular, and radical-containing molecules, in junctions with covalent and donor-acceptor anchor groups, and in cases where the histograms are multi-modal (Figs. 5 and 6).

This theory is based on a physical picture in which the mechanical manipulation of the molecular junction determines the width of the histogram, and the stochastic nature of junction rupture and formation determines its shape. This picture emerged from a recent theory-experiment analysis of the contributing factors to the conductance histogram that showed that this factor alone could account for the width of conductance events encountered in experiments[44].

Equation (8) can be used to understand how molecular and mechanical parameters affect the conductance histograms (Fig. 4 and S2). In particular, we showed that the transmission decay coefficient $\gamma$ determines the conductance width and should be a main parameter to investigate in future works that aim to improve the experimental resolution of conductance measurements. The predictions of Eq. (8) can be used to experimentally test the range of validity of the theory. Moreover, Eq. (8) provides clear targets for atomistic modeling that can be used to computationally recover the conductance distributions as needed to establish contact between simulations and experiments.

We further provided a viable experimental strategy to extract all the microscopic parameters that define the mechanical and conductance properties in the proposed model. For this, it is necessary to complement the conductance histograms with rupture-force or lifetime histograms for the molecular and metal-metal junctions. Such experiments are needed to test the range of applicability of the theory and interpret the conductance histogram in terms of the microscopic parameters defined by the proposed model.

The theory is based on a single well in the free energy surface for the molecular junction. This unimodal contribution to the overall conductance histogram can be isolated using machine-learning clustering of experimental data[7,58,59]. Further, the theory can be generalized to multi-modal processes in which there is interconversion between different junction configurations that can coexist at a given elongation. This physical situation can be represented through multiple wells in the free energy surface in the presence of force.

Overall, the developments in this paper open the possibility to extract physical information from the conductance histograms characterizing the electro-mechanical microscopic properties of molecular junctions, to design meaningful strategies to tune the conductance histogram, and to help bridge the gap between atomistic simulations and molecular junction experiments.

## Methods
### Curve fitting
All fittings were done using the non-linear least squares method, as implemented in the scipy.optimize.curve_fit python package. Example scripts are provided in the Supplementary Information.

### Reporting summary
Further information on research design is available in the Nature Portfolio Reporting Summary linked to this article.

## Data availability
Fitting data is completely defined by the fitting and microscopic parameters reported in Tables 1 and 2.

## Code availability
The script used to fit the conductance histogram is included in the Supplementary Information.

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

## Acknowledgements

We would like to thank Jorge Cossio, Michael Deffner and Carmen Herrmann for helpful discussions. L.M and I.F. thanks the National Science Foundation under Grant No. CHE-1553939 and CHE-2102386 for support. P.C. was supported by MinCiencias, University of Antioquia (Colombia) and the Simons Foundation (USA). The Flatiron Institute is a division of the Simons Foundation.

## Author contributions

L.M. performed all the analyses, simulations and provided all plots. P.C. provided expertize in force-spectroscopy. I.F. developed the hypothesis and supervised the study. All authors discussed the obtained results, and contributed to the theory and writing of the paper.

## Competing interests

The authors declare no competing interests.
