## [Peer Review File · Nature Communications]

REVIEWER COMMENTS

Reviewer #1 (Remarks to the Author):

Break-junction techniques are commonly used in the investigations of single-molecule junctions. Although the conductance histogram is highly reproducible, the conductance of individual molecular junctions is difficult to reproduce in the experiment. This owes to that the structure evolution details of the junctions are hardly to be directly identified and controlled in present experiments [R. Liu, et al, Phys. Rev. Appl., 2018, V9, 054023]. However, the variations of electrode force and conductance versus electrode displacement contain lots of information, which is significantly useful both for identifying the junction structure and for controlling the formation of the junction [Zhou C, et al., Nat. Commun., 2018, V9, 807]. In this paper, the authors developed a theory to fit the conductance histograms by merging the theory of force-spectroscopy with molecular conductance. This theory can capture the shape and peak position of benchmark experimental conductance histograms, and can be used to determine the pulling speed, junction stiffness and temperature. This work is of interest. I could be prepared to recommend the work for publication, but following issues should be further addressed.

(1) The conductance histograms calculated with Eq. 8 in FIG. 3 and FIG.4 show that the right shoulder of the conductance peak is wider than the left shoulder, i.e., the conductance tail spreads towards the higher conductance value. The authors attribute this feature to $\gamma < 0$. However, the break-junction experiments [Zhao Z. et al., Small, 2018, V14, 1703815] show that, for the same type molecules in the same experimental conditions, some conductance histograms show that the tails spread wider on the left than on the right, some spread wider on the right than on the left, while some conductance histograms are almost symmetric. Moreover, the differences in the width for the actual conductance histograms are not as large as FIG 4 (b) shows. This issue should be much better explained in the paper.

(2) The authors partition the break-junction experiment into six main events, and see the metal-metal rupture and molecular junction rupture as two and three brittle springs. Then if the geometry of the electrode tip (not the length of the electrode) is change with the action of the molecule in the elongation of the molecular junction [S. Li, et al., Nano Lett., 2020, V20, 5490; S. S. Wang, et al., J. Phys. Chem. C, 2021, V125, 27290], whether this analogy is still applicable? A short discussion of this issue would be helpful.

(3) (Right column on page 4) Why "Eq. 1 implies that ΔA^\ddagger varies linearly with $F(t)$ "? More explicit descriptions are needed in the context.

(4) A minor point in the last paragraph of right column on page 6: It is "log-transmission", not "transmission".

(5) Why the unit of Free Energy is A and the Elongation is ξ in FIG. 2 ?

Reviewer #2 (Remarks to the Author):

A molecular transport junction is a dynamical system, thus an experiment measuring the conductance of a single-molecule junction involves a large range of molecular geometries. In this manuscript the authors present a microscopic theory of the conductance histograms measured.

The physics of the histogram shapes has been discussed in the literature before (e.g. refs. 54-57), but this work provides an explicit atomistic modeling of the junction rupture occurring in the experiments. This provides a platform for explicit modeling of the histograms. This is a major step forward, and merits publication.

With this new platform specific physical assumptions can now be simulated and discussed:

- it is assumed that a molecular transport junction is brittle. Is this true? Or are the Au atoms so mobile that we are approaching fluid dynamics?

- is this platform necessary? Specifically, are the geometries sampled by a molecule during an experiment caused by the pulling and rupture, as suggested here, or is it the thermal motion of the molecule in the junction? Could these possibilities manifest in separate experimental regimes?

The six stages of the experiment discussed in the theory section must be explicitly marked with the same numbers in figure 1.

In conclusion, this is a landmark paper within theory of molecular transport junctions in particular, and scanning probe experiments in general. I advocate publication.

Reviewer #3 (Remarks to the Author):

The manuscript by L. Mejia et al., provides a method to describe the mechanics of a breaking junction and use this to model the fluctuations due to the mechanical strength at junction formation and rupture and explain the dispersion of conductance measurements of molecular junctions. The model is of interest for the community. Nevertheless, I believe the work presented is not on the level of a publication in nature communications because the model proposed by the authors is quite limited in explaining the fluctuations in molecular junctions and its added value in understanding charge transport through molecular junctions is not yet clear.

My main comments to justify this choice:

- In the third paragraph (part of the motivation of the paper), the authors cite ref 43-47 to explain the spread of conductance and claim a dispersion of 1-2 orders of magnitude. This claim is however not valid thanks to the last 5 years advances in the field. For example, in the reference 7, authors show that using analysis techniques, the conductance value corresponding to a single molecular bridging the gap is much narrower. From the same group, in *Chemical science* 10 (36), 8299-8305 they show that the conductance spread can be explained by different electronic paths through the molecule and in *The Journal of Physical Chemistry C* 126 (20), 8801-8806, similar molecule as the one modelled in the paper, they limit the spread of conductance by separating the traces depending on the molecular configuration. The authors motivate their choice based only on ref 44 (discussing more possibilities to explain the spread in conductance) and do not show a broad knowledge of the main advances in the field.

- The authors have only applied their model to C6DT and C10DT. The authors should apply their model to both conjugated and non-conjugated molecules with different anchoring groups, and display that it generalizes well to all. Several groups have their data open access and they should apply it to more datasets to make a stronger statement. A benchmark study with more data and also AFM measurements where the rupture force can be measured is required to complement this study.

- The modeling is based on the hypothesis provided in the last paragraph on page 2 (hypothesis for the theory part), where the authors argue that the current measurements, which are averages over microsecond, result in the indistinguishability of molecular configurations. This hypothesis is in very strong disagreement with many experimental results, and highly limits the applicability of the model. Besides the papers mentioned earlier, the observation of pi-pi interference between two OPE3 molecules (*Nature Chemistry* volume 8, pages1099–1104 (2016), *J. Am. Chem. Soc.* 2021, 143, 25, 9385–9392) and measurements of conductance modulation in the case of mechano-sensitive molecules (*Nano Lett.* 2018, 18, 9, 5981–5988) show for example that differences in molecular configurations are measurable in the junction. Many of the fundamentally interesting phenomena can only be explained by including conformational effects. The authors should change the claims of the paper before sending it to any other journal. Their approach could be complementary to explain some of the fluctuations but the statement needs to be changed.

Microscopic Theory, Analysis, and Interpretation of Conductance Histograms in Molecular Junctions

Leopoldo Mejía, Pilar Cossio, and Ignacio Franco

Reviewer 1

“Break-junction techniques are commonly used in the investigations of single-molecule junctions. Although the conductance histogram is highly reproducible, the conductance of individual molecular junctions is difficult to reproduce in the experiment. This owes to that the structure evolution details of the junctions are hardly to be directly identified and controlled in present experiments [R. Liu, et al, Phys. Rev. Appl., 2018, V9, 054023]. However, the variations of electrode force and conductance versus electrode displacement contain lots of information, which is significantly useful both for identifying the junction structure and for controlling the formation of the junction [Zhou C, et al., Nat. Commun., 2018, V9, 807]. In this paper, the authors developed a theory to fit the conductance histograms by merging the theory of force-spectroscopy with molecular conductance. This theory can capture the shape and peak position of benchmark experimental conductance histograms, and can be used to determine the pulling speed, junction stiffness and temperature. This work is of interest. I could be prepared to recommend the work for publication, but following issues should be further addressed.”

Author’s reply: We thank the reviewer for carefully considering our manuscript and for the positive assessment of the importance and utility of this work.

“(1) The conductance histograms calculated with Eq. 8 in FIG. 3 and FIG.4 show that the right shoulder of the conductance peak is wider than the left shoulder, i.e., the conductance tail spreads towards the higher conductance value. The authors attribute this feature to $\gamma < 0$. However, the break-junction experiments [Zhao Z. et al., Small, 2018, V14, 1703815] show that, for the same type molecules in the same experimental conditions, some conductance histograms show that the tails spread wider on the left than on the right, some spread wider on the right than on the left, while some conductance histograms are almost symmetric. Moreover, the differences in the width for the actual conductance histograms are not as large as FIG 4 (b) shows. This issue should be much better explained in the paper.”

Author’s reply: We thank the reviewer for bringing this up. Yes, in this model of junction formation and evolution, the direction of the tails of the conductance histogram is associated with the sign of γ . Specifically, for $\gamma < 0$, the tails are to the right as it is statistically more probable to sample the high conductance features of the transmission-vs-elongation behavior of the junction. This model agrees with what is observed in experiments for alkanes (see Fig. 5). The opposite behavior is expected for $\gamma > 0$. More complex transmission-vs-elongation functional relations can lead to other types of behavior. We agree that the differences in the width for the actual conductance histograms are not as large as those in Fig. 4b. In the paper we varied γ over a wide range to exemplify what it does to this contribution of the conductance histogram. In response to the reviewer’s comments we have now more carefully explained this in the paper.

Thanks also for pointing out the paper by Zhao *et al.* which has now been included in the references. In it, the authors develop a scheme to fabricate electrodes that are conical and planar. The main results are in Figure 4 in that paper which show cases, as the reviewer point out, in which the tail goes the left or to the right depending on the molecule and type of electrode used. Note, however, that the experiments have a strong background signal (Fig. 4d) that depends on the value of conductance. Thus, depending on the most probable conductance of the junction the background signal can skew the conductance histogram to the left or to the right. In other words, it is difficult from that figure to determine the intrinsic behavior of the tails of the molecular junction histogram

with respect to the contributions of the noise. A background subtraction will enable us to really determine if the tails are arising from the junction or from the noise. In response to the comments by the reviewer, we have now clarified the need for background subtraction to assess the sign of γ from the histogram.

“(2) The authors partition the break-junction experiment into six main events, and see the metal-metal rupture and molecular junction rupture as two and three brittle springs. Then if the geometry of the electrode tip (not the length of the electrode) is change with the action of the molecule in the elongation of the molecular junction [S. Li, et al., Nano Lett., 2020, V20, 5490; S. S. Wang, et al., J. Phys. Chem. C, 2021, V125, 27290], whether this analogy is still applicable? A short discussion of this issue would be helpful.”

Author’s reply: As the reviewer correctly points out, the geometry of the electrode tip can be deformed as the molecular junction is elongated. For example, for thiol anchor groups on gold, there is evidence that in some instances a string of gold atoms is pulled before the junction breaks (Phys. Rev. Lett. 89, 186402, J. Phys. Chem. C 2015, 119, 27, 15216–15223). This is what in the paper we refer to as plastic deformation of the junction. The model of brittle springs supposes that there is no plastic deformation of the electrodes. However, these events can be captured simply by changing the equilibrium length of the electrodes ξ_{eq} , while leaving the remaining analysis intact. Complementary, by including an additional parameter one can account for more specific deformation properties (see also the second response to Reviewer 2). In response to the comments by the reviewer we have made this aspect more clear in the manuscript.

“(3) (Right column on page 4) Why “Eq. 1 implies that ΔA^\ddagger varies linearly with $F(t)$ ”? More explicit descriptions are needed in the context.”

Author’s reply: In the absence of forces, the spontaneous rate of dissociation is determined by Arrhenius formula $k_0 = N_0 \exp(-\beta \Delta A_0^\ddagger)$ where ΔA_0^\ddagger is the free energy barrier and N_0 the pre-exponential constant. At constant loading rate, the force-dependent free energy profile is $A(\chi, t) = A_0(\chi) + \kappa(\chi - (\dot{F}/\kappa)t)^2/2$, where χ is the reaction coordinate, \dot{F} the loading rate, and κ the spring constant of the cantilever which in the context of molecular junctions will be the elasticity of gold. The system is supposed to be at thermal equilibrium at each instant of time t , and the time label is a convenient way to keep track of the force increase during the experiment. For a soft spring constant,

$$A(\chi, t) \approx A_0(\chi) - \dot{F}\chi t + \frac{(\dot{F}t)^2}{2\kappa}.$$

Now the barrier at time t is

$$\Delta A^\ddagger(t) = A(\chi_{\text{eq}} + \chi^\ddagger, t) - A(\chi_{\text{eq}}, t) = \Delta A_0^\ddagger - \dot{F}\chi^\ddagger t = \Delta A_0^\ddagger - F(t)\chi^\ddagger$$

where we have supposed a constant loading rate and that $F = 0$ at initial time $t = 0$. That is, Bell’s model shows that the free energy barrier linearly decreases with applied force. Thus, in the presence of an applied force the dissociation rate is $k(t) = N_0 \exp(-\beta \Delta A^\ddagger(t)) = k_0 \exp(\beta \chi^\ddagger F(t))$. We have now clarified this in the paper.

“(4) A minor point in the last paragraph of right column on page 6: It is “log-transmission”, not “transmission”.”

Author’s reply: Thanks! This is now fixed.

“(5) Why the unit of Free Energy is A and the Elongation is ξ in FIG. 2 ?”

Author’s reply: Apologies for the confusion. In Fig. 2, A and ξ do not refer to units but to the names of the variables. We have now removed the parenthesis to make this more transparent.

Reviewer 2

“A molecular transport junction is a dynamical system, thus an experiment measuring the conductance of a single-molecule junction involves a large range of molecular geometries. In this manuscript the authors present a microscopic theory of the conductance histograms measured. The physics of the histogram shapes has been discussed in the literature before (e.g. refs. 54-57), but this work provides an explicit atomistic modeling of the junction rupture occurring in the experiments. This provides a platform for explicit modeling of the histograms. This is a major step forward, and merits publication.”

Author’s reply: We thank the reviewer for the positive assessment and for recommending publication.

“With this new platform specific physical assumptions can now be simulated and discussed: - it is assumed that a molecular transport junction is brittle. Is this true? Or are the Au atoms so mobile that we are approaching fluid dynamics?”

Author’s reply: We have so far assumed a simple description of the system’s elongation properties as brittle, which results in tractable analytical expressions of the junction and molecular rupture probabilities. It is possible to introduce an additional parameter in the free-energy profile representation, as is done with μ in Ref. 60, that ranges from brittle to ductile to learn more about the system’s elongation properties. However, this implies having additional parameters to be fitted. We have now mentioned this in the Discussion and left it as a future extension of the work.

“- is this platform necessary? Specifically, are the geometries sampled by a molecule during an experiment caused by the pulling and rupture, as suggested here, or is it the thermal motion of the molecule in the junction? Could these possibilities manifest in separate experimental regimes?”

Author’s reply: Pulling and the stochastic rupture activated by thermal fluctuations are inherent and unavoidable components of the break junction experiments. No matter how sophisticated the experimental technique or analysis is, this stochastic rupture component of the experiment cannot be avoided.

However, this is not the only source of uncertainty in the experiment. The geometry of the junction, binding conformations and thermal events that are slow with respect to the resolution in the current measurement all contribute to the overall conductance broadening. Resolving between all these types of events requires a way to extract further information from the junction. As Reviewer 3 points out, this may be a good place to apply machine learning tools.

Nonetheless, the physical events that we have identified in our work correspond to a major component of the conductance histogram, and by taking them into account it is possible to fit available experiments and extract additional molecular-junction properties. This has been demonstrated in alkanes in the original submission and, in this resubmission, further exemplified in a broader class of molecules. Additional experiments, and theory, are definitively desirable to more deeply understand the relative contributions of the possible physical phenomena at play but, as the reviewer points out, this is “major step forward”.

“The six stages of the experiment discussed in the theory section must be explicitly marked with the same numbers in figure 1.”

Author's reply: Thanks for pointing this out. This has been corrected.

“In conclusion, this is a landmark paper within theory of molecular transport junctions in particular, and scanning probe experiments in general. I advocate publication.”

Author's reply: Thank you very much for carefully reviewing our manuscript and for your support of this contribution.

Reviewer 3

“The manuscript by L. Mejia et al., provides a method to describe the mechanics of a breaking junction and use this to model the fluctuations due to the mechanical strength at junction formation and rupture and explain the dispersion of conductance measurements of molecular junctions. The model is of interest for the community. Nevertheless, I believe the work presented is not on the level of a publication in nature communications because the model proposed by the authors is quite limited in explaining the fluctuations in molecular junctions and its added value in understanding charge transport through molecular junctions is not yet clear.”

Author's reply: Thank you very much for carefully and critically reading our manuscript. From the journal website, the criterion for a paper to be accepted into Nature Communications is that it “should represent an advance in understanding likely to influence thinking in the field, with strong evidence for their conclusions. There should be a discernible reason why the work deserves the visibility of publication in a Nature Portfolio journal.”

We believe our contribution satisfies these criteria. As stated by Reviewer 2, this paper represents a “major step forward” within the theory of molecular junctions in particular and scanning probe experiments in general.

The first advance of this paper is to recognize that even perfect experiments, where the conformation of the electrode and the binding configuration are perfectly well controlled, will still require statistical sampling and exhibit a broad conductance histogram. The width arises because of changes in the conductance during molecular elongation. The need for statistical sampling arises because of the unavoidable thermal stochastic nature of junction rupture events. These insights are supported by the experiments and extensive atomistic simulations of junction formation and evolution in Ref. 44.

The second major advance of this paper is that it summarizes these physical ideas into an analytically tractable and concrete theory for the conductance lineshape that was obtained by merging the theory of single-molecule force spectroscopy with molecular conductance. This novel perspective provides a fresh understanding into the conductance histograms. The validity and relevance of the approach have been demonstrated by showing that the resulting expression for the conductance lineshape can be used to successfully fit the conductance histogram of representative molecular electronic experiments and extract important information. In fact, we now include examples of aliphatic and aromatic molecules and junctions with different anchor groups. This is a significant advance because there is no other physical and interpretable microscopic theory for the conductance lineshape in the literature that can successfully account for these lineshapes. Further, our proposed expressions provide statistically better fits to experiments than existing phenomenological models. For these reasons the results in the paper “represent an advance in understanding likely to influence thinking in the field, with strong evidence for their conclusions”.

With regards to the criterion of why this work “deserves the visibility of publication in a Nature Portfolio journal”, understanding the conductance histogram and designing ways to narrow its width is a central topic in molecular electronics as evidenced by the plethora of references included in our

paper and the papers that have been noted by the reviewers. Notably, systematic progress in this key area has been hindered by the fact that there is no microscopic model that can account for these histograms. In response to this need, this contribution advances the first microscopic theory that can properly account for the lineshapes encountered in experiments and that is testable. This will open opportunities for experiments to augment the information content that can be extracted from the experiments, and to discern between the possible contributing factors that are at play in the experiments. It also opens opportunities to design effective strategies to atomistically model the conductance histograms through free-energy reconstruction techniques.

We stress that the model and theory focus on a particular contribution to the conductance histogram as this approach is necessary to develop a testable and analytically tractable theory. Further generalizing the theory and our understanding of the conductance histograms will require continued efforts by the community. However, the results presented are a significant step forward that will stimulate additional progress.

“My main comments to justify this choice: - In the third paragraph (part of the motivation of the paper), the authors cite ref 43-47 to explain the spread of conductance and claim a dispersion of 1-2 orders of magnitude. This claim is however not valid thanks to the last 5 years advances in the field. For example, in the reference 7, authors show that using analysis techniques, the conductance value corresponding to a single molecular bridging the gap is much narrower. From the same group, in Chemical science 10 (36), 8299-8305 they show that the conductance spread can be explained by different electronic paths through the molecule and in The Journal of Physical Chemistry C 126 (20), 8801-8806, similar molecule as the one modelled in the paper, they limit the spread of conductance by separating the traces depending on the molecular configuration. The authors motivate their choice based only on ref 44 (discussing more possibilities to explain the spread in conductance) and do not show a broad knowledge of the main advances in the field.”

Author’s reply: Thank you for pointing these papers by van der Zant and co-workers out. We are aware of this series of contributions in which unsupervised machine learning is used to cluster conductance data into different classes. What this series of papers show is that the algorithm identifies different binding configurations that can occur during the experiment similar to the example shown in Figure 6 of our paper. In that process, the conductance histogram does get narrower, but it is still quite broad. For instance, Fig. 3 in Ref. 7 shows that the conductance histogram after the clustering shows a dispersion of ~ 1 -2 orders of magnitude in class 2, and of ~ 0.5 -1 orders of magnitude in class 3. The study on porphyrines in Chemical Science shows that after the machine learning clustering analysis class A shows a conductance dispersion of ~ 1 order of magnitude, and ~ 2 orders of magnitude for B and C. Thus, we do not see evidence that this type of analysis overcomes the broad conductance dispersion encountered in experiments. As shown by our analysis the dispersion arises due to changes in conductance during pulling and the unavoidable stochastic nature of junction rupture events. In response to the comments by the reviewer, we have now relaxed our claim to a dispersion of 0.5-2 orders of magnitude to better reflect the progress by the van der Zant group and have included these papers into the manuscript.

In the context of our model, this type of multi-modal processes identified in Figure 6 and also by the machine-learning clustering algorithm, occur when there are multiple wells in the free energy surface when force is applied. Our theory is for the rupture from a single free energy well, but it can be generalized to take into account multiple binding processes as shown in the paper. We have now pointed this out in the outlook.

Importantly, the methods presented in the above references are not in contradiction with our theory. In fact, they are very valuable because they allow to classify the binding modes and narrow the conductance histogram to a unimodal distribution. Thus, there are excellent opportunities for synergy between the two strategies. For instance, we envision using the strategy advanced by van der Zant and co-workers to isolate junctions with specific binding configurations in

force-conductance measurements, and then use the clustered data to extract the electro-mechanical properties of the junction using the proposed model. We have now pointed out these emerging opportunities in the outlook.

“- The authors have only applied their model to C6DT and C10DT. The authors should apply their model to both conjugated and non-conjugated molecules with different anchoring groups, and display that it generalizes well to all. Several groups have their data open access and they should apply it to more datasets to make a stronger statement. A benchmark study with more data and also AFM measurements where the rupture force can be measured is required to complement this study.”

Author’s reply: The model was initially applied to C4SMe, C8SMe, C12SMe, C10DT and C6DT, a total of 5 molecules. However, we agree that all of these cases are aliphatic molecules. In response to the comments by the reviewer, we have now applied the method in 4 aromatic molecules for which we were able to track down data for the conductance histogram. As shown, the fits are excellent demonstrating that the method generalizes well. Overall, this shows that the method can be used in aromatic and aliphatic molecules, and for different types of anchor groups. In response to the comments by the reviewer, we have now added these fits into the paper.

We agree that additional studies with conductance and AFM measurements of the rupture force would be desirable to complement this study. These discussions are currently under way with experimental groups. However, as the reviewer is aware, performing such a class of measurements represent years of additional research efforts and are, thus, outside the scope of the paper. In response to the comments by the reviewer, we have now invited experimental groups to perform measurements in this direction in the outlook of the paper.

“- The modeling is based on the hypothesis provided in the last paragraph on page 2 (hypothesis for the theory part), where the authors argue that the current measurements, which are averages over microsecond, result in the indistinguishability of molecular configurations. This hypothesis is in very strong disagreement with many experimental results, and highly limits the applicability of the model. Besides the papers mentioned earlier, the observation of pi-pi interference between two OPE3 molecules (Nature Chemistry volume 8, pages1099–1104 (2016), J. Am. Chem. Soc. 2021, 143, 25, 9385–9392) and measurements of conductance modulation in the case of mechano-sensitive molecules (Nano Lett. 2018, 18, 9, 5981–5988) show for example that differences in molecular configurations are measurable in the junction. Many of the fundamentally interesting phenomena can only be explained by including conformational effects. The authors should change the claims of the paper before sending it to any other journal. Their approach could be complementary to explain some of the fluctuations but the statement needs to be changed.”

Author’s reply: The time averages of the current is an inherent feature of the physical measurement. These time averages necessarily imply that the experiments cannot measure individual molecular snapshots but the average conductance that is consistent with all molecular conformations that are accessible thermally at a given point in the elongation in about 1 microsecond of measurement.

Very importantly, please note that this does not imply that all molecular conformations will be indistinguishable. All the examples pointed out by the reviewer are cases in which specific conformations are mechanically stabilized or cases in which changes of molecular configuration are slower with respect to the current integration time. In the Nature Chemistry and the Nano Letts articles, the structures are mechanically stabilized. In the JACS, changes in conformation are enacted chemically and do not change during the current measurements.

Other examples from our group include:

1. <https://pubs.acs.org/doi/abs/10.1021/acs.jpcllett.7b03323>. Here gauge conformers in alkanes can be mechanically stabilized and be used to monitor conformational dynamics through conductance. These changes survive the time average inherent to the experiments.

2. <https://pubs.rsc.org/en/content/articlehtml/2017/sc/c8sc04830d> Here we monitor the progress of a chemical reaction through conductance. These changes also survive the time averaging inherent to experiments.

That is, our approach of taking into account time averages is **not** in “very strong disagreement with experimental results”. On the contrary, it reflects a simple but important feature of the experimental conditions and it is consistent with everything that is known in molecular junctions.

In response to the comments by the reviewer, we have now made these aspects more clear in our manuscript.

Overall: We thank the reviewers for their constructive critical comments. We believe that we have carefully and appropriately addressed all the issues that have been raised.

REVIEWER COMMENTS

Reviewer #1 (Remarks to the Author):

I appreciate the effort that the authors put in updating the manuscript. Points 3 to 5 mentioned in my review were addressed adequately. However, points 1 and 2 was not addressed properly. I noticed that the authors gave detailed explanations in their response letter, but the authors should notice that points 1 and 2 need to be discussed explicitly in the manuscript based on the literatures mentioned in the comments. Moreover, in the Reply to the issue 2, the authors said: "The model of brittle springs supposes that there is no plastic deformation of the electrodes." However, the plastic deformations of electrodes have different type. Not all types of plastic deformation must be excluded (e.g. the plastic deformation of the electrode mentioned in [S. S. Wang, et al., J. Phys. Chem. C, 2021, V125, 27290] which does not affect the elastic change of the electrode-molecule distance). Thus, I comment that the second revision to the manuscript is needed.

Reviewer #2 (Remarks to the Author):

The authors have given good and thorough answers to the answers asked by the reviewers. The manuscript emerges even stronger, and I advocate publication.

In my view, this work is of broad interest to people working with scanning probe physics in general and molecular electronics in particular. In my view, a new platform or framework is provided, which has the potential to lift the simulations of molecular transport junctions to the next level. Clearly, a manuscript relevant for Nature Communications.

Reviewer #3 (Remarks to the Author):

I would like to thank the authors for carefully reading the comments. I still believe that the model should be applied to larger dataset of different molecules and techniques (STM BJ or MCBJ for example) to make it more complete as force spectroscopy is not the only parameter that explains the broadening of conductance histograms and make the work more relevant to the community.

Microscopic Theory, Analysis, and Interpretation of Conductance Histograms in Molecular Junctions

Leopoldo Mejía, Pilar Cossio, and Ignacio Franco

Reviewer 1

“I appreciate the effort that the authors put in updating the manuscript. Points 3 to 5 mentioned in my review were addressed adequately. However, points 1 and 2 was not addressed properly. I noticed that the authors gave detailed explanations in their response letter, but the authors should notice that points 1 and 2 need to be discussed explicitly in the manuscript based on the literatures mentioned in the comments. Moreover, in the Reply to the issue 2, the authors said: “The model of brittle springs supposes that there is no plastic deformation of the electrodes.” However, the plastic deformations of electrodes have different type. Not all types of plastic deformation must be excluded (e.g. the plastic deformation of the electrode mentioned in [S. S. Wang, et al., J. Phys. Chem. C, 2021, V125, 27290] which does not affect the elastic change of the electrode-molecule distance). Thus, I comment that the second revision to the manuscript is needed.”

Author’s reply: We thank the reviewer for carefully reading the revised version of this manuscript and for their constructive comments.

With respect to point 1, in the previous version we had already expanded the explanation of how the sign of γ determines the direction of the tails of the conductance histogram. In addition, we added:

“Possible additional asymmetries in the conductance histogram that are introduced by background noise in the experiments need to be removed before inferring the sign of γ from experimental data.”

which is an important discussion point in the context of our manuscript.

In response to the new comments by the reviewer, we have now cited additional references in which the structure of the background noise is discussed. This includes the paper by Zhao pointed about by the referee [Zhao Z. et al., Small, 2018, V14, 1703815], Nano Lett. 2006, 6, 10, 2238–2242 and Phys. Rev. B 84, 205408. The latter two exemplify how to remove the background signal from the conductance data.

We believe that we have now appropriately responded to point 1 by the reviewer, by modifying the paper and including these additional references.

With respect to point 2, the discussion is about the influence of the plastic deformation of the electrodes on the theory. In response to the comments by the reviewer, we had already included the following in the paper:

“At this point the electrodes return to their mechanical equilibrium leaving a gap $\Delta L_f = \xi_0$, where we have assumed that there is no plastic deformation of the gold electrodes, i.e. changes in the electrode geometry due to the mechanical manipulation. Such events change the equilibrium length of the electrodes but leave the analysis intact. Further extensions of the model that allow transitions from brittle to ductile regimes can be captured by adding an additional parameter to the free energy profile, as proposed in Ref. [60].”

In response to the new comments by the reviewer, to better reflect the literature, we have now added the following references, including the J.Phys.Chem.C suggested by the reviewer and one of our contributions in this direction discussing possible plastic deformations of the junction:

<https://journals.aps.org/prl/abstract/10.1103/PhysRevLett.89.186402>

<https://doi.org/10.1021/acs.jpcc.5b02843>

<https://pubs.acs.org/doi/abs/10.1021/acs.jpcc.1c07935>

<https://pubs.acs.org/doi/abs/10.1021/acs.jctc.2c00648>

The reviewer suggest that we explicitly distinguish between different types of plastic deformations of the electrodes and that we discuss them in the paper. We have incorporated the elements of this discussion that are relevant to our paper and have now represented the possible mechanisms accurately through references. A more detailed discussion in the manuscript of these mechanisms will

reduce the clarity of the paper as it will ask the reader to change focus from the “theory of conductance histograms” to “different mechanisms for plastic deformation of metal electrodes”.

We also note that the NanoLett previously suggested by the reviewer <https://pubs.acs.org/doi/abs/10.1021/acs.nanolett.0c02015>, while an interesting contribution, it is focused on the creation of direct Ag-C metal bonds which is not the topic of this paper. For this reason, we do not see how to include this paper in the discussion in a way that it improves the quality of our manuscript.

Reviewer 2

“The authors have given good and thorough answers to the answers asked by the reviewers. The manuscript emerges even stronger, and I advocate publication.”

In my view, this work is of broad interest to people working with scanning probe physics in general and molecular electronics in particular. In my view, a new platform or framework is provided, which has the potential to lift the simulations of molecular transport junctions to the next level. Clearly, a manuscript relevant for Nature Communications.”

Author’s reply: We thank the reviewer for carefully reading through our manuscript and for advocating publication in Nature Communications.

Reviewer 3

“I would like to thank the authors for carefully reading the comments. I still believe that the model should be applied to larger dataset of different molecules and techniques (STM BJ or MCBJ for example) to make it more complete as force spectroscopy is not the only parameter that explains the broadening of conductance histograms and make the work more relevant to the community.”

Author’s reply: We thank the reviewer for this constructive critical comment. We agree with the reviewer that testing the theory in a wider data set is desirable. However, we had exhausted representative data sets that were readily available in the literature as, unfortunately, there is no repository for molecular conductance histograms.

As suggested by the reviewer in a subsequent communication, we contacted directly Prof. van der Zant, Prof. Venkataraman, and Prof. Hong. All of these authors kindly provided additional data sets. The van der Zant group provided benchmark data on alkane dithiols and diamines measured through the Mechanically Controlled Break Junction (MCBJ) technique.¹ Prof. Venkataraman on molecules containing radicals,² and Prof. Hong on metallofullerene complexes³ both measured with the STM BJ technique.

We have now successfully fitted all these histograms using our proposed formula for the lineshape, underscoring the general utility of the approach. In response to the comments by the reviewer, we have augmented Fig. 5 and Table 2 in the paper to include this additional data, and added Fig. S5 and Table S2 in the supplementary information.

We have decided to keep the fits for the data obtained using the MCBJ in the Supplementary Information (Fig. S5 and Table S2). Our formula fits very well these conductance histograms and even the narrower histograms isolated by machine learning, suggesting that the functional form for MCBJ and STM-BJ is the same. However, in the MCBJ experiments the geometry of the force is more complex as the applied force bends the nanojunction in the three-point configuration generating an elongation of the electrode gap. That is, in MCBJ the applied force is not parallel to the junction elongation (see Rev. Sci. Instrum. 82, 053907 (2011), now cited in the paper), while in our model the elongation and force are collinear. For this reason, we do not expect the microscopic parameters needed to define the histograms for the two techniques to be identical as, contrary to the geometry of our model and the STM-BJ, in MCBJ the applied force is not parallel to the junction elongation. In

this light, we decided to include this data in the Supplementary Information and added the following discussion in the main manuscript:

“Figures 5 and 6 are based on STM break-junction (STM-BJ) measurements. Another experimental setup often used to construct conductance histograms is the mechanically-controlled break-junction setup (MCBJ) in a three-point bending configuration.⁴ For completeness, we have included fits using Eq.8 of MCBJ experiments performed on alkanedithiols and alkanediamines¹ in Fig. S5 and Table S2 of the supplementary information. The excellent fits suggest that the functional form is also applicable in this case. However, we do not expect the microscopic parameters needed to define the MCBJ to be identical to those of our model and the STM-BJ, as in the MCBJ the applied force is not parallel to the junction elongation.”

Developing a microscopic theory that matches the geometry of the MCBJ is outside the scope of this single paper.

Overall: We thank the reviewers for their constructive critical comments. We believe that we have carefully and appropriately addressed all the issues that have been raised.

References

- ¹ Frederik H Van Veen, Luca Ornago, Herre SJ Van Der Zant, and Maria El Abbassi. Benchmark study of alkane molecular chains. *J. Phys. Chem. C*, 126(20):8801–8806, 2022.
- ² Liang Li, Jonathan Z Low, Jan Wilhelm, Guanming Liao, Suman Gunasekaran, Claudia R Prindle, Rachel L Starr, Dorothea Golze, Colin Nuckolls, Michael L Steigerwald, et al. Highly conducting single-molecule topological insulators based on mono-and di-radical cations. *Nat. Chem.*, 14(9):1061–1067, 2022.
- ³ Jing Li, Songjun Hou, Yang-Rong Yao, Chengyang Zhang, Qingqing Wu, Hai-Chuan Wang, Hwei Zhang, Xinyuan Liu, Chun Tang, Mengxi Wei, et al. Room-temperature logic-in-memory operations in single-metallofullerene devices. *Nat. Mater.*, 21(8):917–923, 2022.
- ⁴ Christian A Martin, Roel HM Smit, Ruud van Egmond, Herre SJ van der Zant, and Jan M van Ruitenbeek. A versatile low-temperature setup for the electrical characterization of single-molecule junctions. *Rev. Sci. Instrum.*, 82(5), 2011.

REVIEWERS' COMMENTS

Reviewer #1 (Remarks to the Author):

The authors have carefully responded to the comments. The manuscript has been improved and acceptable.

Reviewer #3 (Remarks to the Author):

I would like to thank the authors for the extra effort in benchmarking their model. The new figures and results are very convincing and therefore, I would suggest to publish this article.

Microscopic Theory, Analysis, and Interpretation of Conductance Histograms in Molecular Junctions

Leopoldo Mejía, Pilar Cossio, and Ignacio Franco

Reviewer 1

“The authors have carefully responded to the comments. The manuscript has been improved and acceptable.”

Author’s reply: We thank the reviewer for their input during the reviewing phase. We do believe that the paper has improved as a result of this process.

Reviewer 3

“I would like to thank the authors for the extra effort in benchmarking their model. The new figures and results are very convincing and therefore, I would suggest to publish this article.”

Author’s reply: We thank the reviewer for the suggestions and constructive comments. They have tremendously helped us improve this article.